

# False head complexity and evidence of predator attacks in male and female hairstreak butterflies (Lepidoptera: Theclinae: Eumaeini) from Mexico

Eric Novelo Galicia[1], Moisés Armando Luis Martínez[2] and Carlos Cordero[3]

[1] Facultad de Ciencias, Universidad Nacional Autónoma de México, Ciudad de México, Mexico
[2] Museo de Zoología "Alfonso L Herrera", Departamento de Biología Evolutiva, Facultad de Ciencias, Universidad Nacional Autónoma de México, Ciudad de México, Mexico
[3] Departamento de Ecología Evolutiva, Instituto de Ecología, Universidad Nacional Autónoma de México, Ciudad de México, México

Corresponding author
Carlos Cordero,
cordero@ecologia.unam.mx,
crafaelcm@gmail.com

## ABSTRACT

In many butterfly species, the posterior end of the hindwings of individuals perching with their wings closed resembles a butterfly head. This "false head" pattern is considered an adaptation to deflect predator attacks to less vulnerable parts of the body. The presence of symmetrical damage in left and right wings is considered evidence of failed predator attacks to perching butterflies. In this research, we tested the prediction derived from the deflection hypothesis that the degree of resemblance of the false head area (FH) to a real head, as measured by the number of FH "components" (eyespots, "false antennae", modified outline of the FH area and lines converging on the FH area) present in the hindwings, is positively correlated to the frequency of symmetrical damage in the FH area. We studied specimens from two scientific collections of butterflies of the subfamily Theclinae (Lycaenidae) belonging to the Universidad Nacional Autónoma de México (Colección Nacional de Insectos [CNIN] and Museo de Zoología, Facultad de Ciencias [MZFC]). We scored the presence of symmetrical damage in a sample of 20,709 specimens (CNIN: 3,722; MZFC: 16,987) from 126 species (CNIN: 78 species; MZFC: 117 species; 71 species shared by both collections) whose hindwings vary in the number of FH components, and found that, as predicted, the proportion of specimens with symmetrical damage increases as the number of FH components increases. We also tested the hypothesis that behavioural differences between the sexes makes males more prone to receive predator attacks and, thus, we predicted a higher frequency of symmetrical damage in the FH of males than in that of females. We found that the frequency of symmetrical damage was not significantly different between males and females, suggesting that behavioural differences between the sexes produce no differences in the risk of being attacked. Overall, our results provide support to the idea that the FH of butterflies is an adaptation that deflects predator attacks to less vulnerable parts of the body in both sexes.

## INTRODUCTION

Predation is a major selective pressure responsible for numerous prey adaptations (*Wickler, 1968*; *Ruxton, Sherratt & Speed, 2004*). In butterflies, visually oriented predators have selected for several components of the colour pattern and morphology of the wings (*Robbins, 1980*; *Howse, 2014*; *Rossato et al., 2018*). The deflection of predator attacks to "expendable" parts of the body is a type of defence (*Cooper Jr, 1998*; *Humphreys & Ruxton, 2018*) that appears to be present in several butterfly species in which the posterior end of the hindwings resembles a butterfly head (at least to the human eye) when the butterfly perches with its wings closed (Fig. 1; *Wickler, 1968*; *Robbins, 1980*; *Cordero, 2001*; *Ruxton, Sherratt & Speed, 2004*). The 'false head' (FH hereafter) is composed of a number of wing pattern and wing shape elements (*Robbins, 1980*; *Cordero, 2001*) that vary between species and produce different degrees of similarity to a real head. The four FH components identified by *Robbins (1981)* (and slightly modified by us) are the following: (1) presence of colour patterns in the posterior end of the hindwings resembling eyes (= eyespots; Fig. 1: *Pseudolycaena damo*). (2) Presence of tails resembling antennae in the posterior end of the hindwings (Fig. 1). (3) Outline of the posterior end of the hindwings modified in a way that resembles the contour of a head (Fig. 1: both species, but especially clear in *Pseudolycaena damo*). (4) Presence, in the ventral side of both wings, of lines converging towards the posterior end of the hindwings (Fig. 1: *Arawacus sito*; Fig. 2: *Micandra cyda* and *Laothus erybathis*); these lines presumably direct the "predator's eye" to the FH area. FHs are particularly frequent in species belonging to the subfamily Theclinae (Lycaenidae) (*Robbins, 1980*) and behaviours possibly enhancing deception by the FH have been identified (*Robbins, 1980*; *Cordero, 2001*).

Although there are observations of predators directing their attacks to the butterfly FH (*Van Someren, 1922*; *Sourakov, 2013*; *López-Palafox & Cordero, 2017*), published reports are scant. Further support for the deflection function of FHs comes from a recent study in which salticid spiders were exposed to virtual prey (resembling insects) on a computer screen (*Bartos & Minias, 2016*). The spiders preferred to attack the head of the virtual prey, but they were induced to increasingly attack the rear end as the experimenters added FH components (i.e., eyespots, tails resembling antennae, etc.) to the virtual prey. On the other hand, indirect evidence supporting the deflection hypothesis comes from two studies documenting symmetrical damage in the FH area, a likely result of failed predator attacks (*Robbins, 1981*; *Tonner et al., 1993*). Direct observations confirm symmetrical damage in the FH area of both hindwings resulting from predator attacks (*Van Someren (1922)* observations of lizard attacks).

*Robbins (1981)* reasoned that if the number of FH components in the wings of a butterfly species influences its degree of deceptiveness for predators, then that number should be directly proportional to the fraction of individuals exhibiting symmetrical damage in the FH area. This author analyzed rates of symmetrical damage in the FH area of Neotropical species of the tribe Eumaeini (Lycaenidae: Theclinae) that vary in the number of FH components. He used two samples, one of "more than 1,000 specimens of about 125 species" collected over a period of 5 weeks in an area of 0.5 km² in Colombia, and another of "almost 400

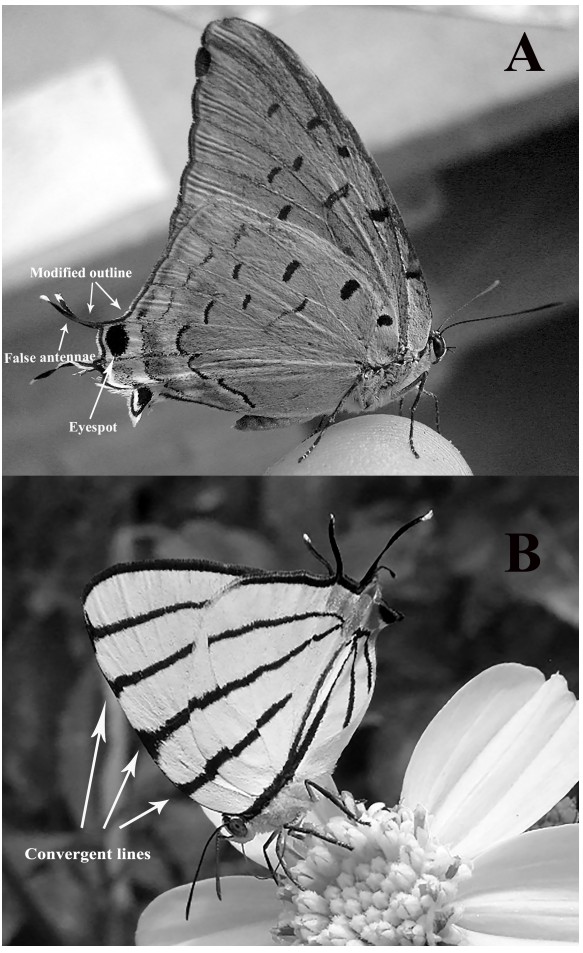

**Figure 1** **Two examples of "false heads" as seen in perching Lycaenidae butterflies.** (A) *P. damo*. (B) *A. sito*. False head components are indicated with arrows. Photo credit: Juan Carlos García Morales.

specimens of about 75 species" collected over a period of three months in Panama, and found support for his hypothesis. Here, we first provide further tests of Robbins' hypothesis by analyzing two samples of Lycaenidae with characteristics different from those used by him. We used specimens belonging to two scientific collections, which were captured by different collectors in several localities covering many parts of Mexico, along a period of several decades. Secondly, since in many Lycaenidae species males employ a sit-and-wait mate location strategy, perching in consecutive days in the same places, at the same time of the day and returning to the same perches after inspecting flying objects (*Johnson & Borgo, 1976*; *Alcock, 1983*; *Cordero & Soberon, 1990*; *Fischer & Fiedler, 2001*; *Takeuchi & Imafuku, 2005a*; *Takeuchi & Imafuku, 2005b*; *Salazar, 2011*; *Dinesh & Venkatesha, 2013*), we also explored the hypothesis that males, due to their stationary, conspicuous and predictable behaviour, suffer higher rates of predator attacks than females.

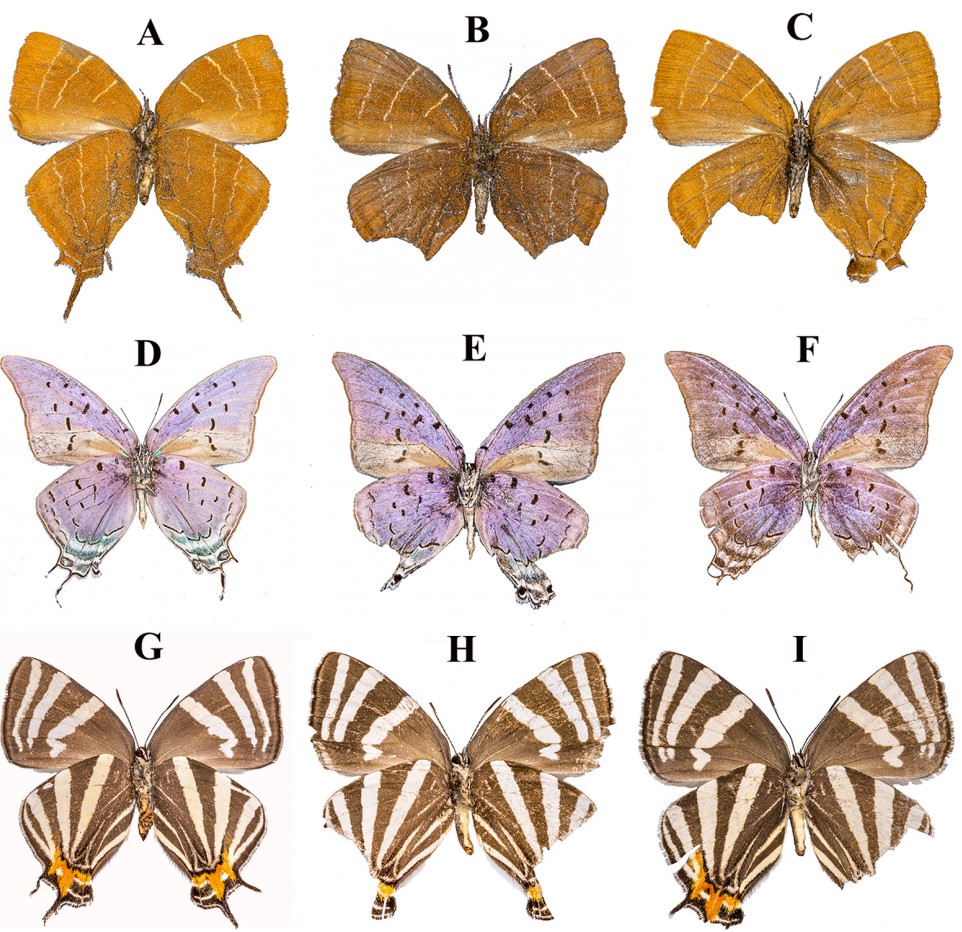

**Figure 2   Examples of the ventral wing surfaces of pinned specimens of "false head" butterfly species (Lepidoptera: Lycaenidae: Theclinae).** (A–C) *Micandra cyda* (♀). (D–F) *Pseudolycaena damo* (♂). (G–I) *Laothus erybathis* (♀). (A) (D) (G) specimens with almost intact hindwings. (B) (E) (H) specimens with symmetrical damage in the false head area (considered evidence of deflected predator attacks). (C) (F) (I) specimens with non-symmetrical damage in the false head area. Photo credit: Raúl Iván Martínez.

## MATERIALS AND METHODS

### Scoring evidence of predator attacks and number of false head components in collection specimens

We studied pinned specimens from two scientific collections of Mexican Theclinae (Lycaenidae) belonging to the Universidad Nacional Autónoma de México. The first sample is part of the Colección Nacional de Insectos of the Instituto de Biología (hereafter CNIN), and the second sample is part of the collection of butterflies of the Museo de Zoología "Alfonso L. Herrera" of the Facultad de Ciencias (hereafter MZFC).

To assess the hypothesis that the fraction of individuals exhibiting symmetrical damage in the FH area is directly proportional to the number of FH components, we examined a total of 20,709 specimens (CNIN: 3,722; MZFC: 16,987) of 126 species (CNIN: 78 species; MZFC: 117 species; 71 of these species were present in both collections) belonging to 52

genera (species names according to *Robbins, 2004*). We only included in our study species with at least 10 specimens. We recorded the presence of symmetrical damage (i.e., very similar in both hindwings) in the FH area (Fig. 2). We did not consider symmetrical damage: (a) the loss of all or most of both hindwings, (b) damage in the FH area that had a different shape in each hindwing (Fig. 2), and (c) damage in the FH area resulting from aging (i.e., ragged wings). We scored, for each species, the number of FH components described in the first paragraph of the introduction (Fig. 1), briefly: (1) eyespots, (2) tails ("false antennae"), (3) modified outline of the posterior end of the hindwings, and (4) presence of lines converging in the FH area in the ventral side of the wings. The number of FH components was used as an estimate of FH similarity to a real head or FH "complexity". To test the hypothesis that the proportion of specimens exhibiting symmetrical damage in the FH area is larger in males than in females, we used the sample of species in which the sex of the specimens was recorded in the labels (all 78 species from the CNIN; Appendix S1) or was determined by us (75 of the 117 species from the MZFC; Appendix S1).

## Statistical analyses

We used generalized linear mixed models (GLMMs) with Poisson error distribution and log link function to evaluate the effect of the number of FH components on the probability of specimens having symmetrical damage. Since a phylogeny, including all species in our sample, is not available, we first performed analyses considering each species as an independent data point. We support this decision on *Robbins (1981)* evidence that FH components evolve rapidly. However, in an attempt to consider a possible role of phylogenetic inertia (*Blomberg & Garland Jr, 2002*), we also ran the analyses at the genus level (using average numbers of FH components and of specimens with symmetrical damage). In a second set of models using the subset of species (or genera) for which sex was determined, the effect of sex and the interaction of sex and the number of FH components were also included.

In all of the models, the response variable was the number of individuals with symmetrical damage; the logarithm of the total specimens was included as an offset to convert this to a proportion (*Hilbe, 2011*). The fixed effect of main interest was the number of FH components, and the collection of origin (MSFC or CNIN) was included as a fixed effect in all models to account for differences in the overall level of damage between collections (e.g., due to differences in collection locations, seasons, collectors). In the two models in which sex and the interaction between sex and number of FH components were considered, they were included as fixed effects. Finally, since proportions of specimens with symmetrical damage were calculated separately by collection in the first set of models and by sex and collection in the second set, there were multiple measures of most species or genera. Species was therefore included as a random intercept effect in species level models and genus as a random intercept in genus level models to account for these multiple measures.

We simplified initial models via backwards-stepwise simplification, in which the effects of variables on the model were tested by removing the variables in order from least- to most significant *P*-value, comparing these nested models using ANOVA, and removing variables whose removal does not provoke a significant effect, until only significant variables

remain (resulting in the "minimum adequate model"; *Crawley, 2013*). We also evaluated the models using the Akaike Information Criterion (AIC) with similar results (shown in Appendix S2). We generated all statistical analyses and graphs in the R statistical software, version 3.5.1 (*R Development Core Team, 2016*) using the RStudio interface, version 1.1.456 (*R Studio Team, 2015*). GLMMs were constructed using the "lme4" package, version 1.1-20 (*Bates et al., 2014*). Graphs were generated using the package "sciplot" (*Murdoch, 2017*). Model diagnostics (homogeneity of variance and normality of residuals, and lack of over/underdispersion, outliers, and zero-inflation) were verified by inspection of simulated residuals using the "DHARMa" package (*Hartig, 2019*) and were fulfilled by all models. We include the R code used in these models in Appendix S3.

# RESULTS

## General observations

In the whole sample, the percentage of specimens with symmetrical damage in the FH area was 1.21% (N = 251 = CNIN: 32 + MZFC: 219). In the sub-sample of specimens whose sex was recorded, the proportion of specimens with symmetrical damage in the FH area was similar to that in the whole sample, 1.31% (N = 173 = CNIN: 32 + MZFC: 141). The mean percentage of individuals with symmetrical damage per species was small in both collections: 1.02% (median = 0%, $Q_{25}$%–$Q_{75}$% = 0–0.7%, minimum–maximum = 0–10%) in the CNIN and 1.38% (median = 0.15 0.12%, $Q_{25}$%–$Q_{75}$% = 0–0%, minimum–maximum = 0–15.4%) in the MZFC. The percentage of species with at least one specimen with symmetrical damage was almost twice as large in the MZFC collection (59 species, 50.4% of the sample) than in the CNIN collection (21 species, 26.3% of the sample) (Chi-squared = 10.69, P = 0.0011, *df* = 1).

## Effect of number of FH components on evidence of attacks in the FH area

As expected, in both the analyses at the species and at the genus level, we found highly significant positive effects of the number of FH components on the proportions of specimens with symmetrical damage in the FH area (Table 1, Figs. 3A and 3B). Analyses at both levels also detected an effect of the collection of origin of the data on the proportions of specimens with symmetrical damage (Table 1): these proportions were significantly larger in collection MZFC than in collection CNIN (Figs. 3A and 3B).

## Effect of sex of the specimens on evidence of attacks in the FH area

The data did not support the prediction that the percentage of individuals with symmetrical damage in the FH area would be larger in males than in females. Neither in the analyses at the species level nor at the genus level, the sex of the specimens or the interaction between sex and number of FH components had a significant effect on the proportion of specimens with symmetrical damage (Table 2). As in the previous section, in the sample of species in which the sex of the specimens was known, at both the species and genus level, a significant and positive effect of the number of FH components on the proportion of specimens with symmetrical damage was found (Table 2). However, in contrast to the

**Table 1** **Results of models evaluating the effects of the number of false head (FH) components and the collection of origin (MZFC or CNIN) on the probability of symmetrical damage at the species (a) and genus (b) level.** In both cases, both the number of FH components and collection of origin had significant effects, such that the initial model is the minimum adequate model. Significant $P$-values are in bold type.

| Fixed Effects | $\beta \pm SE$[a] | $Z$[a] | $P_Z$[a] | $P_{\chi 2}$[b] |
|---|---|---|---|---|
| (a) Species level | | | | |
| Number of FH components | $0.6291 \pm 0.1701$ | 3.757 | **$1.72 \times 10^{-4}$** | **$3.103 \times 10^{-4}$** |
| Collection (MZFC-CNIN) | $0.4000 \pm 0.1952$ | 2.049 | **0.0405** | **0.0325** |
| Random effects: $\sigma^2_{species} = 0.6987$ $n = 197$ total observations from 124 species | | | | |
| (b) Genus level | | | | |
| Number of FH components | $0.7465 \pm 0.2329$ | 3.205 | **$1.35 \times 10^{-3}$** | **$4.006 \times 10^{-3}$** |
| Collection (MZFC-CNIN) | $0.5168 \pm 0.1937$ | 2.668 | **$7.64 \times 10^{-3}$** | **$6.95 \times 10^{-3}$** |
| Random effects: $\sigma^2_{genus} = 0.4494$ $n = 81$ total observations from 49 genera | | | | |

**Notes.**
[a] Parameters from model.
[b] $P$-value resulting from $\chi^2$ nested model comparisons following removal of each variable during backwards stepwise simplification.

previous section, the effect of the collection of origin of the data on the proportion of specimens with symmetrical damage was significant only in the genus level analysis, where a larger proportion of specimens with symmetrical damage was also detected in the MZFC collection (Table 2).

# DISCUSSION

The shapes and colours that make butterfly wings aesthetically appealing to humans are, in large part, products of the evolutionary pressures imposed by predators (*Howse, 2014*; *Rossato et al., 2018*). The "false head" of Lycaenidae is considered an adaptation for deflecting predator attacks to less vulnerable areas of the animal. *Robbins (1980)* found that the complexity of FHs is positively associated with the proportion of specimens with symmetrical damage in the FH area, a likely evidence of failed predator attacks (*Van Someren, 1922*). Here, using larger and more heterogeneous samples, we confirm that the probability of finding symmetrical damage in the FH area is larger in species and genera with more FH components (Tables 1 and 2, Fig. 3), providing further evidence for the role of predator deflection in the evolution of butterfly false heads and for the idea that FHs with more components are better at deceiving predators.

In contrast, our hypothesis that the sit-and-wait mate location strategy employed by males of many Lycaenidae species (*Johnson & Borgo, 1976*; *Alcock, 1983*; *Cordero & Soberon, 1990*; *Fischer & Fiedler, 2001*; *Takeuchi & Imafuku, 2005a*; *Takeuchi & Imafuku, 2005b*; *Salazar, 2011*; *Dinesh & Venkatesha, 2013*) results in males being more stationary and behaviourally conspicuous than females, and thus more prone to be attacked by visually oriented predators, was not supported by the data. Males and females did not differ in the proportion of specimens with symmetrical damage (Table 2). It is possible that our generalization is wrong and the number of species in which males exhibit sit-and-wait mate location behaviour is much smaller than we think. If this is the case, the prediction

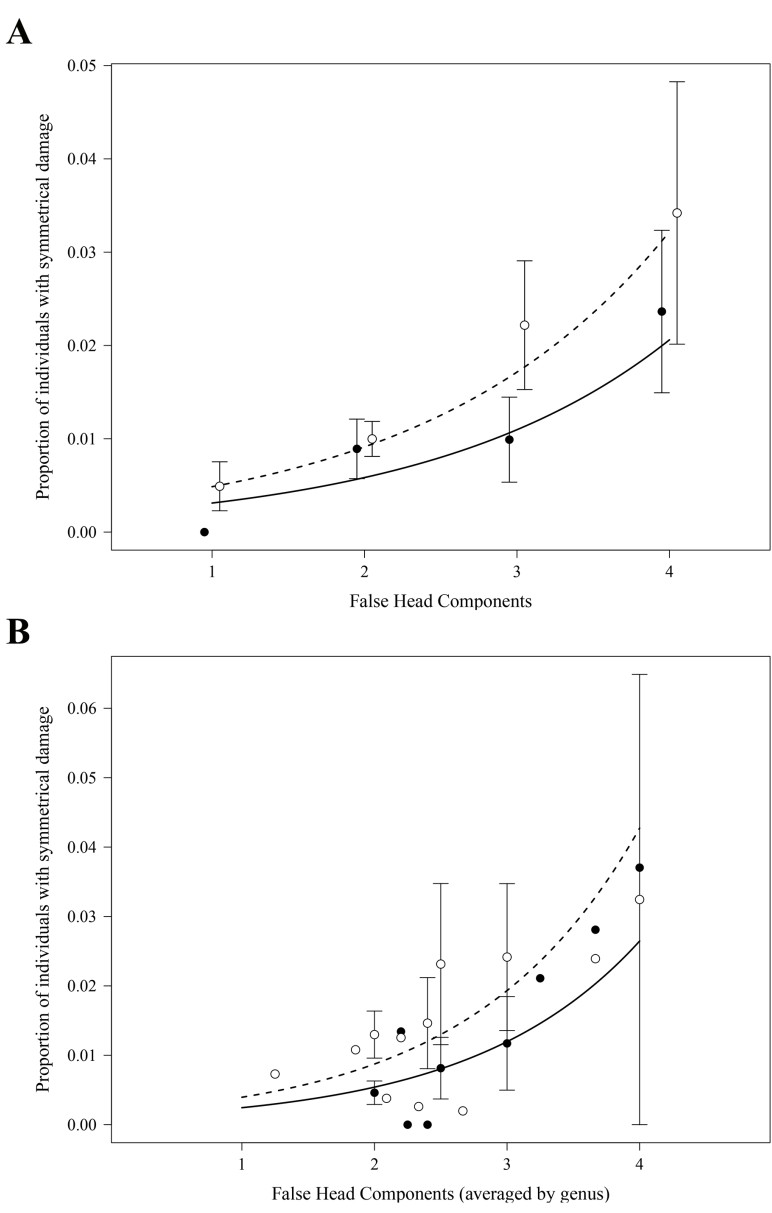

**Figure 3** **Effect of the number of false head components on the proportion of individuals per species (A) and per genus (B) with symmetrical damage.** Proportions were calculated separately for each of the two scientific collections where specimens were examined. Each point represents the average per species (A) or genus (B) within each collection (closed points: CNIN, open points: MZFC); error bars show standard error. Curves show the predicted proportion of individuals with damage based on negative binomial generalized linear models, accounting for the significant additive effect of collection (solid line: CNIN, dashed line: MZFC). At the genus level (B), the average of the number of FH components per genus was used.

that males will suffer higher rates of attack should be tested only in species exhibiting sit-and-wait male mate location strategy. Furthermore, female behaviour, especially egg laying and nectar feeding, could expose females to a predation risk similar to that experienced by

**Table 2  Results of models evaluating the effects of sex and the interaction of sex and number of false head (FH) components, in addition to number of FH components and collection of origin (MZFC or CNIN), on the probability of symmetrical damage at the species (a) and genus (b) level.** In both models, sex and the interaction between sex and number of FH components were non-significant and therefore were removed from the initial models (a, c), leaving number of FH components and collection of origin as the only variables in the minimum adequate models (b, d). Significant *P*-values are in bold type.

| Fixed Effects | $\beta \pm SE$[a] | $Z$[a] | $P_Z$[a] | $P_{\chi 2}$[b] |
|---|---|---|---|---|
| (a) Species level initial (complete) model | | | | |
| Number of FH components | $0.7459 \pm 0.2299$ | 3.244 | **$1.18 \times 10^{-3}$** | **$7.652 \times 10^{-3}$** |
| Collection (MZFC-CNIN) | $0.3989 \pm 0.2129$ | 1.874 | 0.06095 | 0.055 |
| Sex | $1.1220 \pm 0.6293$ | 1.783 | 0.07459 | 0.8001 |
| Number of FH components * Sex | $-0.3851 \pm 0.2145$ | $-1.795$ | 0.07259 | 0.0724 |
| Random effects: $\sigma^2_{species} = 0.5674$ $n = 306$ total observations from 105 species | | | | |
| (b) Species level minimum adequate model | | | | |
| Number of FH components | $0.4999 \pm 0.1827$ | 2.736 | **$6.21 \times 10^{-3}$** | – |
| Collection (MZFC-CNIN) | $0.3955 \pm 0.2126$ | 1.860 | 0.06282 | – |
| Random effects: $\sigma^2_{species} = 0.5478$ $n = 306$ total observations from 105 species | | | | |
| c) Genus level initial (complete) model | | | | |
| Number of FH components | $0.6274 \pm 0.2840$ | 2.209 | **0.0272** | **0.0488** |
| Collection (MZFC-CNIN) | $0.3999 \pm 0.2035$ | 1.966 | **0.0493** | **0.0424** |
| Sex | $0.6065 \pm 0.6769$ | 0.896 | 0.3703 | 0.8300 |
| Number of FH components * Sex | $-0.2165 \pm 0.2479$ | $-0.873$ | 0.3825 | 0.3842 |
| Random effects: $\sigma^2_{genus} = 0.4399$ $n = 150$ total observations from 47 genera | | | | |
| (d) Genus level minimum adequate model | | | | |
| Number of FH components | $0.4936\ 0.2415$ | 2.044 | **0.041** | – |
| Collection (MZFC-CNIN) | $0.3989\ 0.2035$ | 1.960 | **0.050** | – |
| Random effects: $\sigma^2_{genus} = 0.4404$ $n = 150$ total observations from 47 genera | | | | |

**Notes.**

[a]Parameters from model.

[b]*P*-value resulting from $\chi^2$ nested model comparisons following removal of each variable during backwards stepwise simplification.

males. In fact, the lack of obvious sexual differences in FH traits observed in our samples also point against our hypothesis.

If we accept the conclusion that a FH with all four components is better at deflecting predator attacks, then we need to understand why species with different numbers of FH components frequently coexist in the same location (e.g., *Robbins, 1981*; *López-Palafox, Luis-Martínez & Cordero, 2015*). *Robbins (1981)* considered that genetic restrictions or ecological factors could be responsible for explaining variation in FH complexity. *Robbins (1981)* considers unlikely that genetic restrictions provide a general explanation, because evidence indicates that FH components evolve rapidly. Our data are consistent with this point of view: 11 of the 26 genera (42%) with more than one species in our database have species belonging to two (8 genera) or three (3 genera) different categories of number of FH components (Table 3). Thus, ecological factors could provide better explanations for the diversity observed in FH patterns. Species with different numbers of FH components could represent alternative adaptive solutions to selective pressures from visually oriented predators when we consider additional traits. For example, a species with a "complete" FH

**Table 3  Genera in which species with different numbers of false head (FH) components were observed.** Subscript numbers in parenthesis indicate the number of species with that number of FH components in our sample (see Appendix S1 for species names).

| Genus | Number of FH components in different species |
|---|---|
| 1. *Arawacus* | $3_{(1)}, 4_{(2)}$ |
| 2. *Atlides* | $2_{(4)}, 3_{(1)}$ |
| 3. *Contrafacia* | $2_{(1)}, 3_{(1)}$ |
| 4. *Cyanophrys* | $1_{(1)}, 2_{(6)}$ |
| 5. *Erora* | $1_{(2)}, 2_{(2)}$ |
| 6. *Rekoa* | $2_{(3)}, 3_{(2)}$ |
| 7. *Theritas* | $2_{(2)}, 3_{(2)}$ |
| 8. *Tmolus* | $2_{(1)}, 3_{(2)}$ |
| 9. *Ministrymon* | $1_{(1)}, 2_{(2)}, 3_{(3)}$ |
| 10. *Panthiades* | $2_{(1)}, 3_{(1)}, 4_{(2)}$ |
| 11. *Strymon* | $1_{(2)}, 2_{(6)}, 3_{(3)}$ |

could be rather conspicuous when perched (Fig. 1), whereas a species like *Callophrys xami* with only eyespots and tails (*López-Palafox & Cordero, 2017*), but with the ventral side of the wings of green colour, could be hard to spot when posed on or near leaves, as well as being able to deflect predator attacks to the FH once it is detected. The morphological components of the FH considered in this paper are complemented with behaviours that possibly enhance the deceptive effect (*Robbins, 1980*; *Cordero, 2001*; see also *Hoskins & Bálint, 2016*), such as the back-and-forth movement of the hindwings along the sagittal plane frequently performed while resting by Theclinae species (*López-Palafox & Cordero, 2017*) and a few species of other groups (Riodinidae, *Robbins, 1985*; Nymphalidae, *Sourakov, 2015*). The interaction of these behavioural traits with the morphological components could also help explain the interspecific variation in the complexity of the FH.

One of the referees of this paper (Dr. A. Krupitsky) noted the interesting fact that in some species the degree of development or even the number of some of the FH components varies between individuals (the reviewer mentions *Cyanophrys* spp., *Micandra cyda* and *Oenomaus ortygnus* among these species). We did not consider this possibility in our study and, thus, we did not look for intraspecific variation in the FH components. In our opinion, quantitative documentation of the extent of this variation and its distribution between the different taxa, as well as its possible relationships with the average number of FH components and sex (among other variables), is an important subject for future research. Speculate about the possible causes of such intraspecific variation seems premature, although relaxed selection from reduced predation pressure comes to mind as one possible hypothesis. Additionally, species with substantial intraspecific variation could be used as "natural treatments" in the functional study of FH components, in experiments similar to that of *López-Palafox & Cordero (2017)* on "false antennae".

## CONCLUSIONS

The false head (FH) hindwing pattern is considered an adaptation to deflect predator attacks away from the real head to less vulnerable parts of the body. We tested one prediction of this hypothesis using two large samples of museum specimens. Since FHs vary between species in the number of morphological and colour components (''complexity''), we predicted a positive correlation between the number of FH components and the proportion of specimens with evidence of predator attacks directed to the FH area. We found the predicted association whether we used species or genus as a sample unit. We also tested the related hypothesis that behavioural differences between the sexes makes males more prone than females to receive predator attacks. Contrary to our prediction, we found that the proportion of specimens with evidence of predator attacks directed to the FH area was not different between males and females. Overall, our results support the deflection hypothesis and the idea that FHs with more elements are better at deceiving predators, and suggest that behavioural differences between the sexes produce no differences in the risk of being attacked.

## ACKNOWLEDGEMENTS

We thank Drs. Jorge Llorente and Alejandro Zaldívar for allowing us to study the specimens from the collections of the Museo de Zoología ''Alfonso L. Herrera'' (Facultad de Ciencias, UNAM) and the Colección Nacional de Insectos (Instituto de Biología, UNAM), respectively. We thank Dr. Lynna Kiere for her critical help with the statistical analyses. We thank Juan Carlos García Morales for allowing us to use his excellent photographs in Fig. 1. We thank the thoughtful commentaries of Drs. Zsolt Bálint, Anatoly Krupitsky and Andrei Sourakov on a previous version of the manuscript. We thank Isabel Vargas, María Cristina Mayorga and Raúl Iván Martínez for superb technical support.

### Funding
This work was supported by PAPIIT/UNAM (México) under grant IN210715 to Carlos Cordero. Eric Novelo was supported by a scholarship from PAPIIT/UNAM. The funders had no role in study design, data collection and analysis, decision to publish, or preparation of the manuscript.

### Grant Disclosures
The following grant information was disclosed by the authors:
PAPIIT/UNAM (México): IN210715.
PAPIIT/UNAM.

### Competing Interests
The authors declare there are no competing interests.

## Author Contributions

- Eric Novelo Galicia performed the experiments, analyzed the data, prepared figures and/or tables, authored or reviewed drafts of the paper, approved the final draft.
- Moisés Armando Luis Martínez performed the experiments, contributed reagents/materials/analysis tools, authored or reviewed drafts of the paper, approved the final draft, confirmed the identification of the butterfly species.
- Carlos Cordero conceived and designed the experiments, analyzed the data, contributed reagents/materials/analysis tools, prepared figures and/or tables, authored or reviewed drafts of the paper, approved the final draft.

## Data Availability

The raw data are available in Appendix S1. The code used for the statistical analyses is available in Appendix S3.

## Supplemental Information

Supplemental information for this article can be found online at http://dx.doi.org/10.7717/peerj.7143#supplemental-information.

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
