# Peer review of "False head complexity and evidence of predator attacks in male and female hairstreak butterflies (Lepidoptera: Theclinae: Eumaeini) from Mexico"

_PeerJ, doi:10.7717/peerj.7143_

## Round 0.1 · original submission · Minor Revisions

Dear Dr. Novelo and colleagues:

Thanks for submitting your manuscript to PeerJ. I have now received three independent reviews of your work, and as you will see, the reviewers raised some minor concerns about the research. Despite this, these reviewers are optimistic about your work and the potential impact it will have on research studying predator avoidance in Lycaenidae butterflies.

Therefore, I am recommending that you revise your manuscript accordingly, taking into account all of the issues raised by the reviewers. I do believe that your manuscript will be ready for publication once these issues are addressed.

Good luck with your revision,

-joe

·

Basic reporting

no comment

Experimental design

no comment

Validity of the findings

no comment

Additional comments

This paper expands upon previous collections-based studies of functional morphology of the false head (FH) wing pattern elements in Lycaenidae butterflies. The study is well executed and is a valuable addition to our understanding of wing pattern evolution in Lycaenidae. I believe, it can be published in its present form.
The study can be used as a template for similar studies of FH (and other attack-deflecting wing pattern elements) that are quite common in Lepidoptera. There are many examples of such elements. For example, Tonner et al. (1993) demonstrated by analyzing wing damage in Stichophthalma camadeva (Nymphalidae) that about a third of butterflies in their sample bore signs of presumed predator attacks in FH region of the wing. In another nymphalid, Archaeoprepona chromus, the FH pattern elements are enhanced by sagittal movements of the hindwings, which are initiated when approaching predator is detected (Sourakov 2015). This behavior is akin to that displayed by the hairstreaks. Authors may want to include this in the Discussion section of their paper.

Tonner, M., Novotny, V., Leps, J., Komarek, S. 1993. False head wing pattern of the Burmese junglequeen butterfly and the deception of avian predators. Biotropica 474-478.

Sourakov, A., 2015. Antipredation and “antimimicry”: wing pattern is supported by behavior in Archaeoprepona chromus (Lepidoptera: Nymphalidae: Preponini). ATL Notes, December 2015 issue: 1-7.

·

Basic reporting

no comment

Experimental design

see the general comments to the authors

Validity of the findings

see the general comments to the authors

Additional comments

The topic is interesting as it tackles a subject which is connected to the subfamily Theclinae. Thecline imagines possess a peculiar wing shape and ventral wing pattern posing the false-head hypothesis. This is briefly but well posed in the introductory part.

The paper is written in a clear and unambiguous language. The manuscript was composed with high care. There are very few typos and all the references-citations are relevant. Figures are in high quality, and all necessary, the legends are informative enough. The descriptions and the data supplied are enough to reproduce the experiment.

The authors examined more than 20 thousand well curated specimens for their experiment. The working hypothesis they posed are simple but important enough for having the further step in understanding the role of the false-head pattern in theclines. The statistical method for me seems to be relevant and well chosen (although I am not an expert in biostatistics).

I have some remarks and notes which are listed below, I ask the authors for taking these notes into consideration.

1. Title should be more precise: False head complexity and evidence of predator attacks in male and female hairstreak butterflies (Lepidoptera: Theclinae: Eumaeini) in the Neotropical region; the subject of the paper are hairstreaks imagines form Mexico (I think the single copper species should be taken out from the species examined (see later below)
2. Line 29 – The acronyms of CNIN and MZFC in the abstract are unresolved, it is impossible or difficult to understand them for the reader who is not familiar in the usage museum acronyms.
3. Lines 53-57 – The elements of the false head are listed. Although I absolutely understood the description but I am sure that readers less familiar in butterfly wing pattern these lines could pose some difficulties. Why you do not give an extra figure for showing these components, or use Fig. 1 to mark them with arrows pointing to the component?
4. Paragraph starts from line 73 – It is necessary to cite Robbins 1981 only once. Please remark that Robbins analysed only the Neotropical hairstreaks.
5. Line 103 “belonging tp 49 genera” – Please indicate what kind of nomenclature and taxonomy is used, I guess Robbins 2004 in Lamas.
6. Line 109 – Again, the false-head components should be shown visually in a figure.
7. Line 116 – probably at the end of this paragraph a reference should be given to the Supplementary Material.
8. Line 122 – probably it would be good here to mention that Robbins and his colleagues shown that the male androconia is also the subject of rapid evolution (there are papers on Arcas, Denivia and Thereus).
9. Lines 122-125 In fact… - This sentence should better fit under the Results or Discussion, and probably it would be necessary to show these results in more detailed manner (for example listing and discussing the genera, the phenomenon….); but this is not essential for the paper.
10. Line 137: “sex*” – why the asterisk?
11. Line 148 “results (not shown)” – why these results are not shown? if they are really VERY similar, this should be stated more strongly proving it is not necessary to show the details, but if they are just somewhat similar…
12. Lines 185-187: “However…” – this statement is not clear enough just like this, more explanation would be welcome.
13. Lines 190-198: “The shapes and…. induced by the FH)” – This is just a repetition of the introduction.; should be deleted.
14. Line 204: write Theclinae, instead of the typo “Techlinae”; but probably it would be better to write “Lycaenidae” (see the next entry).
15. Lines 205 – the references Fischer & Fiedler and Dinesh & Venkatesh do not refer to Theclinae; see above the entry no. 15.
16. Line 210 – the semi sentence “if this is…” should be not in brackets; it is an important remark.
17. Line 222 – I would be happy to see here this result also in a Table with the names of the genera.
18. Line 231 – FH can be supplemented with special behaviour (see Bálint & Hoskins, Lepidoptera Novae 9: 33-36 (2016)
19. Figure 1 _ probably it would be important also to present an image what show the false head from a different view – most probably more relevant for the paper; this is what I mean: https://www.butterfliesofamerica.com/L/imagehtmls/LycRio/ADW-Atlides_h_halesus_F_Fruit_Cove_St_Johns_Co_FL_USA_30-X-05_2_i.htm
20. Figure 2. – please write that the images show the ventral wingsurfaces; please give scales (indicating forewing lengths or giving scale bars); the specimens of Laothus erybathis seem to be all females (on the basis of wingshape, abdominal tip; and the male have a blue reflector and lacks the V shaped medial pattern in cell CuA2)
21. All the following remarks concern to the Supplementary material
- why the FH component are not independently coded? that would be more easy to check the evaluation? Because of this I do not really understand some of the values
- for having more clarity it would be better to list the taxa in alphabetical order according to genera, and not according to the collections;
- Iophanus pyrrhias is not a thecline species, it is a Lycaeninae; should be removed from the list;
- I suggests the following corrections, changes
1. Brangas jetus = Brangas getus
2. Bussa busa = Brevianta busa
3. Laspis castimonia = Iaspis castimonia
4. Symbiosis (???) = probably Symbiopsis
5. Thecla arza = Arzecla arza
6. Thecla caninius = Gargina caninius
7. Thecla ligurina = Kolana ligurina
8. Thecla lyde = Kolana lyde
9. Thecla phobe = Nicolaea phobe
10. Thecla seton = Arzecla sethon
11. Thecla thoria = Gargina thoria

·

Basic reporting

English used throughout the manuscript is clear and technically correct (except one point: I suggest more widely used term “[wing] margin” instead of “outline” in lines 26, 55, 110).
Figures are relevant to the content of the article, of sufficient resolution, and appropriately described and labeled. Additionally, I suggest to include scheme (or colour photo) of the hindwing of species with four false head elements numbered and marked with arrows for better understanding.
I thank you for providing the raw data. I suggest to include geographical labels data of the taxa in question in the table (if possible). Also, please check line 11 of the table (delete the species name).

Experimental design

No comment.

Validity of the findings

Discussion of FH elements in morphologically variable species should be included and appropriately stated in conclusion (please see “General comments for the author” section).

Additional comments

I commend the authors for their extensive dataset and solid background based on both publications of predecessors and previous works of the authors of the current manuscript, and I’m generally satisfied with presentation of the data, their interpretations, and conclusions. The research in question fits into the broader field of knowledge devoted to prey-predator interactions.
One point was not considered in the manuscript: how does the intraspecific morphological variation (degree of FH elements development) occuring in some species (e.g., Micandra cyda, Oenomaus ortygnus, Cyanophrys species, etc.), and widely known in Eumaeini in the whole, impact on the statistical analysis and general conclusions? Hindwings of some treated species vary in the number of FH components (eyespots and lines converging towards the posterior end of the hindwings) due to individual variations. I suggest this point to be added in the discussion and conclusion.

---

## Round 0.2 · accepted · Accept

Dear Dr. Novelo and colleagues:

Thanks for revising your manuscript based on the minor concerns raised by the reviewers. I now believe that your manuscript is suitable for publication. Congratulations! I look forward to seeing this work in print, and I anticipate it being an important resource for research communities studying predator avoidance in Lycaenidae butterflies. Thanks again for choosing PeerJ to publish such important work.

Best,

-joe

#